Direct measurement forest carbon protocol: a commercial system-of-systems to incentivize forest restoration and management

Marino Bruno D.V. bruno.marino@pem-carbon.com 1
Truong Vinh 2
Munger J. William 3
Gyimah Richard 4
1 Executive Management, Planetary Emissions Management Inc. , Cambridge , MA , United States of America
2 Planetary Emissions Management Inc. , Cambridge , MA , United States of America
3 School of Engineering and Applied Sciences and Department of Earth and Planetary Sciences, Harvard University , Cambridge , MA , United States of America
4 Forestry Commision of Ghana , Accra , Ghana, Africa
Giordani Paolo
Electronic publication date: 2020 Apr 27
Publication date: 2020
Volume: 8
Electronic Location ID: e8891
Received 2019 Oct 1; Accepted 2020 Mar 11
Copyright: ©2020 Marino et al.
Copyright year: 2020
Copyright holder: Marino et al.
License: This is an open access article distributed under the terms of the Creative Commons Attribution License, which permits unrestricted use, distribution, reproduction and adaptation in any medium and for any purpose provided that it is properly attributed. For attribution, the original author(s), title, publication source (PeerJ) and either DOI or URL of the article must be cited.
License URL: https://creativecommons.org/licenses/by/4.0/

Keywords: Harvard forest, Ankasa park ghana, Forest carbon quantification, Forest carbon trading, Deforestation, Forest net ecosystem exchange, Paris agreement, REDD+, Climate action reserve, Clean development mechanism

Funding: The authors received no funding for this work.

==============================
Forest carbon sequestration offsets are methodologically uncertain, comprise a minor component of carbon markets and do not effectively slow deforestation. The objective of this study is to describe a commercial scale in situ measurement approach for determination of net forest carbon sequestration projects, the Direct Measurement Forest Carbon Protocol™, to address forest carbon market uncertainties. In contrast to protocols that rely on limited forest mensuration, growth simulation and exclusion of CO2 data, the Direct Measurement Forest Carbon Protocol™ is based on standardized methods for direct determination of net ecosystem exchange (NEE) of CO2 employing eddy covariance, a meteorological approach integrating forest carbon fluxes. NEE is used here as the basis for quantifying the first of its kind carbon financial products. The DMFCP differentiates physical, project and financial carbon within a System-of-Systems™ (SoS) network architecture. SoS sensor nodes, the Global Monitoring Platform™ (GMP), housing analyzers for CO2 isotopologues (e.g., 12CO2,13CO2, 14CO2) and greenhouse gases are deployed across the project landscape. The SoS standardizes and automates GMP measurement, uncertainty and reporting functions creating diverse forest carbon portfolios while reducing cost and investment risk in alignment with modern portfolio theory. To illustrate SoS field deployment and operation, published annual NEE data for a tropical (Ankasa Park, Ghana, Africa) and a deciduous forest (Harvard Forest, Petersham, MA, USA) are used to forecast carbon revenue. Carbon pricing scenarios are combined with historical in situ NEE annual time-series to extrapolate pre-tax revenue for each project applied to 100,000 acres (40,469 hectares) of surrounding land. Based on carbon pricing of $5 to $36 per ton CO2 equivalent (tCO2eq) and observed NEE sequestration rates of 0.48 to 15.60 tCO2eq acre−1 yr−1, pre-tax cash flows ranging from $230,000 to $16,380,000 across project time-series are calculated, up to 5×  revenue for contemporary voluntary offsets, demonstrating new economic incentives to reverse deforestation. The SoS concept of operation and architecture, with engineering development, can be extended to diverse gas species across terrestrial, aquatic and oceanic ecosystems, harmonizing voluntary and compliance market products worldwide to assist in the management of global warming. The Direct Measurement Forest Carbon Protocol reduces risk of invalidation intrinsic to estimation-based protocols such as the Climate Action Reserve and the Clean Development Mechanism that do not observe molecular CO2 to calibrate financial products. Multinational policy applications such as the Paris Agreement and the United Nations Reducing Emissions from Deforestation and Degradation, constrained by Kyoto Protocol era processes, will benefit from NEE measurement avoiding unsupported claims of emission reduction, fraud, and forest conservation policy failure.

Introduction

Forest landowners and forest communities typically lack economic incentives and social benefits to balance deforestation with conservation and preservation (Duguma et al., 2019). A constellation of factors is responsible for deforestation (Busch & Ferretti-Gallon, 2017), claiming ∼50% of tropical forested landscapes (Brancalion et al., 2019; Rozendaal et al., 2019), including contested land rights, high carbon project cost and requirements for landowners (Kerchner & Keeton, 2015), failure of payment for ecosystem services (Fenichel et al., 2018; Lamb et al., 2019), low or negative payments resulting from the United Nations Reducing Emissions from Deforestation and Degradation (REDD+) programs (Köhl, Neupane & Mundhenk, 2020), and as we argue here, uncertainty for forest carbon sequestration (Engel et al., 2015; Marino, Mincheva & Doucett, 2019; Zhang, 2019). Carbon markets are primarily driven by reduction/avoidance of emissions to the atmosphere from energy production and consumption (Liddle, 2018) while investment in removal of CO2 from the atmosphere by reforestation and conservation has not gained carbon market traction (Gren & Zeleke, 2016; Laurance, 2007) declining by ∼72% from 2011 to 2016 (Hamrick & Gallant, 2017; Molly Peters-Stanley Gonzalez & Yin, 2013). Discount pricing for forest carbon (e.g., <$5 tCO2eq1, 2017: <1$, 2018) (Hamrick & Gallant, 2018; Hamrick & Gallant, 2017) results in limited ecological, social and economic benefits of carbon trading to stakeholders due, in part, from risk of offset invalidation intrinsic to estimation protocols.

Estimation protocols do not directly observe forest CO2 fluxes; terms for ecosystem photosynthesis and respiration are absent unavoidably introducing uncertainty for annual net forest carbon determination and monetization to carbon markets (Dunlop, Winner & Smith, 2019; Haya, 2019; Marino, Mincheva & Doucett, 2019). Invalidation risk for estimation protocols stems from reliance on forest mensuration (e.g., timber survey) conducted every 6 or 12 years (California Air Resources Board, 2015a; Forest Carbon Partners, 2013; Marland et al., 2017) coupled with tree growth simulation models to infer annual changes in net forest carbon offsets (California Air Resources Board, 2011; California Air Resources Board, 2014; California Air Resources Board, 2015b; Climate Action Reserve, 2018). Forest mensuration uncertainties up to 80% for individual trees, and up to 20% for plot level estimation of annual forest carbon have been reported (Gonçalves et al., 2017; Holdaway et al., 2014; Paré et al., 2015) suggesting that such uncertainty is unaccounted for in estimation based protocols.

Estimation based protocols including the California Air Resources Board (CARB) (California Air Resources Board, 2014), the Climate Action Reserve (CAR) (Climate Action Reserve, 2018), the American Carbon Registry (ACR) (Winrock International, 2016), the Verified Carbon Standard (VERRA), and the Clean Development Mechanism (CDM) (Zhang et al., 2018) share offset uncertainty (Köhl, Neupane & Mundhenk, 2020; Kollmuss & Fussler, 2015) in the absence of direct measurement. Approximately 0.9 billion hectares of forests are available worldwide for large-scale restoration opportunities (Bastin, 2019; Brancalion et al., 2019), however, in addition to carbon quantification uncertainties, financing for large-scale projects has proven difficult (Foss, 2018). Complete and direct carbon accounting of forests is required to track biospheric carbon dynamics given the limited and impermanent nature of forest and soil carbon (Baldocchi & Penuelas, 2019; Schlesinger, 2003; Schlesinger & Amundson, 2018). Here, we address forest carbon accounting uncertainties by linking direct measurement of net ecosystem exchange (NEE) of forest carbon fluxes for a project with carbon market transactions in a Direct Measurement Forest Carbon Protocol (DMFCP).

Figure 1 Showing an overview of the DMFCP structure and process.

The Direct Measurement Forest Carbon Protocol (DMFCP) measures gross vertical fluxes of carbon forest ecosystems important for carbon trading shown as: geographical project boundary (dashed line); NEE, net ecosystem exchange of CO2 fluxes; AGC, above ground carbon; BGC, below ground carbon; Photosynthesis, the total carbon uptake by plants or gross primary productivity (GPP); Respiration of ecosystem (Reco), total sources of CO2 released to the atmosphere from plants (AGC, Ra) and soil microbes (BGC, Rh); SoS sensor network; and, a Global Monitoring Platform (GMP). The SoS network and GMP’s are deployed across the project landscape, according to an engineering plan specifying number, height and placement of sensors, to determine net ecosystem exchange (NEE) representing net forest carbon sequestration for a project. Forest carbon gross fluxes (GPP, Reco) measured in situ and resulting in NEE is designated as Physical Carbon, total land area and time period of project performance are designated as Project Carbon, and annual accounting and registration of project carbon provides the basis (e.g., quantity of tCO2eq available) and pricing for sale of Financial Carbon. Multiple projects and resulting forest carbon products are combined in a Pooled Portfolio and listed in a registry detailing project accounting and verification criteria. Pooled Portfolio carbon products, based on equivalent carbon accounting, can be sold to voluntary and compliance buyers worldwide. Pooled Portfolio products may also incorporate additional greenhouse gase fluxes (e.g., CH4, N2O) and isotopic forms (e.g., isotopologues)2 that can be measured with precision in the field and typically reported in the delta notation with per mil unilts. 3The geographical project boundary may be comprised of local, regional or larger land areas (e.g., state, country). Project types include: R, reforestation refers to a project that plants trees on a site previously forested; AD, avoided deforestation refers to a project that prevents deforestation; FM, forest management refers to a project that improves the net carbon sequestration; AF, afforestation refers to a project that establishes trees on land that otherwise would not be planted; AG, agroforestry refers to a project that combines forest conservation and or tree planting with agriculture; TM, timber/wood products involves sustainable harvest of timber within the project area resulting in wood products for construction and manufacturing. Traditional protocols do not directly observe CO2 but rely on proxies and estimation. The DMFCP is formalized with standardized intake forms listing a project (e.g., project listing application) and a project management plan defining terms and conditions for carbon product operations across multiple 10-year intervals. NEE records reductions in photosynthesis caused by fire and deforestation should these events occur in the project areas. Standing carbon inventory derived from biometric or remote sensing methods will be employed to augment and cross-check project NEE data. The SoS and GMPs operate as an integrated autonomous system to monitor, measure and transform GHG flux data relative to local, regional and global reference materials for bulk and isotopic composition, providing the basis for calculation of verified tradeable GHG financial products that differentiate biogenic from anthropogenic net GHG fluxes (Marino, 2013; Marino, 2014a; Marino, 2017b; Marino, 2017a; Marino, 2014b; Marino, 2014c; Marino, 2015b; Marino, 2015a; Marino, 2016b; Marino, 2016d; Marino, 2016c; Marino, 2016a; Marino, 2019).

The objective of the DMFCP is to efficiently monetize sustainable forest management and direct revenue to landowners and communities in lieu of deforestation. The DMFCP commercializes large-scale (e.g., 1 + million hectares), direct, in-situ measurement of vertical gross forest CO2 fluxes (e.g., photosynthesis and ecosystem respiration) to determine net forest carbon sequestration or net ecosystem exchange (NEE), a universal feature of NEE research platforms (Baldocchi, 2019; Baldocchi, Chu & Reichstein, 2018; Baldocchi & Penuelas, 2019; Burba, 2013). The DMFCP, employing a network system architecture, the SoS, and a sensor platform, the GMP, account for carbon from measurement-to-monetization of NEE based products as described in Fig. 1 (overview) and Fig. 2 (annual accounting). NEE has been measured in over 600 locations worldwide (Fluxdata, 2020a; Novick et al., 2018) but has not been utilized to support commercial SoS networks for realization of verified forest carbon products and carbon market transactions. NEE, notwithstanding limitations intrinsic to the methodology, offers a transformative advancement compared to estimation protocols for annual net forest carbon sequestration that lack direct CO2 measurement (e.g., gC m−2 yr−1). The DMFCP commercial platform is described employing NEE data from two research sites, the Ankasa Park tropical rainforest located in Ghana, Africa (Nicolini, 2012), and the Harvard Forest deciduous forest site located in Petersham, MA, USA (Barford et al., 2001; Munger, 2016; Urbanski et al., 2007). The NEE time series data for each site, in combination with carbon pricing scenarios, is used to establish revenue projections across an areal expanse of 100,000 acres (404,685.6 hectares). We compare landowner benefits and incentives to restore forests and reverse deforestation employing the DMFCP and traditional estimation-based protocols as well as compare uncertainties for each approach and their significance to supporting verifiable forest carbon financial products.

Methods

Net ecosystem exchange (NEE) is a measure of the net exchange of carbon fluxes between an ecosystem and the atmosphere (per unit ground area) and is a universally accepted and fundamental metric of ecosystem carbon sink strength (Baldocchi, Chu & Reichstein, 2018; Baldocchi & Penuelas, 2019; Kramer et al., 2002). NEE can be defined as: (1) NEE=GPP+Reco

and, (2) Reco=Ra+Rh,

where GPP = gross primary production or photosynthetic assimilation, Reco = ecosystem respiration, Ra = autotrophic respiration by plants, and Rh = heterotrophic respiration by soil microbes. NEE can be expressed as Net Ecosystem Productivity (NEP) plus sources and sinks for CO2 that do not involve conversion to or from organic carbon: − NEE = NEP + inorganic sinks for CO2 − inorganic sources of CO2 (Chapin et al., 2006; Lovett, Cole & Pace, 2006; Luyssaert et al., 2009). NEE measurements integrate (1) and (2) (e.g., Burba, 2013), consistent with the focus presented here on sequestration and monetization of biospheric carbon where CO2 reduction/increase is a credit/debit to forest and biospheric carbon storage. For example, a negative NEE flux represents a net carbon sink into the biosphere (e.g., removal or capture of CO2 from the atmosphere) and a positive NEE represents a net carbon source into the atmosphere from the biosphere (e.g., increase of CO2 in the atmosphere). The sign convention accommodates the definition of a carbon credit as representing 1 tone CO2 equivalent (CO2eq)ii sequestered or captured from the atmosphere (Kollmuss et al., 2010). We assume that loss of carbon due to fire, UV, removal of biomass and import of biomass is negligible as both project sites are protected (Ankasa Park) or managed as conserved land (Harvard Forest). NEE potentially records reductions in photosynthesis caused by fire and deforestation should these events occur in the project areas (Goulden et al., 2006; Mamkin et al., 2019; Ney et al., 2019; Ueyama et al., 2019). Standing live carbon inventory derived from biometric and or remote sensing methods, typically would be employed to augment NEE data (Ouimette et al., 2018; Verma et al., 2013).

Figure 2 Showing DMFCP components and time series.

(A) Features and benefits of the DMFCP comprised of the SoS and GMPs include: (1) direct field measurement of NEE (CO2, CH4 and N2O) relative to a zero-emission baseline showing positive, negative or neutral GHG emission employing an SoS and one or more GMPs; a positive NEE indicates a GHG source or emission to the atmosphere from the biosphere, whereas a negative NEE indicates a CO2 sink or emission reduction from the atmosphere and results in carbon credits or offsets, (2) ex ante, annual accreditation periods that can be applied to multiple GHG’s, (3) exits from a landowner agreement after 10 years with a penalty according to a ton-year accounting calculation, (4) landowner benefit from initial upfront payment (t0) and annual royalty on sales payment (t1). (B) Multiple projects subsequent to data quality checks by a data center can be listed in a registry and grouped into pooled portfolios; verification of system performance by external third-party verifiers of reference values and calibration of GHG analyzers is performed according to operation of the SoS. Products can be purchased by voluntary and compliance buyers worldwide through multiple sales channels. The hypothetical values shown for CO2, CH4 and N2O (bars) resulting from a field sensor platform are negative in year one and mixed in years 2 and 10. Simple addition of the values for each GHG for annual periods result in a positive, negative or neutral GHG balance. Multiple projects located in specified property boundaries can be grouped to address simple numerical additionality. The DMFCP process simplifies existing protocols for forest carbon sequestration (Table 1). Traditional protocols rely on proxies for CO2 (i.e., not measured or observed at any time in the protocol process) to establish a baseline and test for additionality. NEE records reductions in photosynthesis caused by fire and deforestation should these events occur in the project areas. Standing carbon inventory derived from biometric or remote sensing methods will be employed to augment and cross-check project NEE data. The SoS and GMPs operate as an integrated autonomous system to monitor, measure and transform GHG flux data relative to local, regional and global reference materials for bulk and isotopic composition, providing the basis for calculation of verified tradeable GHG financial products that differentiate biogenic from anthropogenic net GHG fluxes (Marino, 2013; Marino, 2014a; Marino, 2017b; Marino, 2017a; Marino, 2014b; Marino, 2014c; Marino, 2015b; Marino, 2015a; Marino, 2016b; Marino, 2016d; Marino, 2016c; Marino, 2016a; Marino, 2019).

A detailed review of the eddy covariance method, parameter values, data processing codes and uncertainties for determination of NEE are not under study in this work and have been reported in detail elsewhere (Blackwell, Honaker & King, 2017; Vitale, Bilancia & Papale, 2019). A data processing flowchart for Fluxnet2015, the source of data used in this study, is available (https://fluxnet.fluxdata.org/data/fluxnet2015-dataset/data-processing/). Uncertainties of up to 20% of annual flux for a single NEE tower have been reported (Finkelstein & Sims, 2001) but vary according to random and systematic uncertainty terms and applicable corrections. Annual mean NEE fluxes between co-located towers were found to be within 5% of each other for the Howland Forest eddy covariance site (Hollinger & Richardson, 2005). NEE uncertainty can be influenced by the number of towers for a given project area (Kessomkiat et al., 2013), instrument noise, spectral attenuation, atmospheric turbulence, and data processing (Foken, Aubinet & Leuning, 2012; Polonik et al., 2019), in addition to upscaling from tower footprint to larger areas (Hollinger & Richardson, 2005; Richardson et al., 2006; Mauder et al., 2013). NEE uncertainties for single and multiple towers are actively under study as are quantitative corrections including those for: (1) chronic underreporting of nocturnal fluxes due to low friction velocity (Aubinet, Vesla & Papale, 2012; (Staebler & Fitzjarrald, 2004; Wutzler et al., 2018), (2) filtering of raw EC data for conversion to half-hourly data commonly reported and as reported in this study (Fluxnet, 2020b), (3) gap-filling protocols (Reichstein et al., 2012), and, (4) extrapolation of EC data to areas outside of the EC footprint (e.g., single sites and networks) (Jung et al., 2019; Wang et al., 2016). The reported corrections and uncertainties are noted for the data employed in this study.

NEE data for a single tower for each site was accessed from online data sources and transformed into tones carbon dioxide equivalent per acre per year (e.g., tCO2eq acre−1 yr−1) (Supplemental Information 2). The NEE values for both sites representing footprints of ∼1–10 km2 are used to extrapolate NEE to 100,000 acres (40,469 hectares) to illustrate potential revenue for large-scale projects. The extrapolation of NEE data is for illustration purposes only as single tower data for both sites may not be representative of larger forest areas, discussed below. Extrapolated NEE values were combined with carbon prices ranging from $5 to $36 tCO2eq to explore pre-tax revenue scenarios including definition of hypothetical carbon products underlying the projections. Cumulative tCO2eq is based on summing the annual tCO2eq for each record across the extrapolated area of 100,000 acres (40,469 hectares).

Field Sites

Ankasa Park, Ghana, Africa (Fig. 3A)

The Ankasa Park (AP) eddy covariance tower (5°17′00″N2°39′00″W: GH-Ank) is located in a wet evergreen forest in south-western Ghana (Fig. 3A) (Nicolini, 2012) within the Ankasa Conservation Area. The 62-meter-high AP tower equipped with an open path CO2 analyzer was developed and operated as part of the CarboAfrica project (Stefani et al., 2009) and was operational for four years (2011 to 2014) by the University of Tuschia, Italy (Nicolini, 2012). NEE data used in this analysis are available online from the Fluxnet (2015) quality-checked database (http://fluxnet.fluxdata.org/data/fluxnet2015-dataset/) as annual NEE based on the gap filled VUT_NEE_REF values (e.g., Wutzler et al., 2018). The NEE data are gap-filled, filtered and corrected for low friction velocity periods that likely underestimate night time respiration (“Data Processing—Fluxdata, 2020a; ORNLDAAC, 0000; Nicolini, 2012). Uncertainty for the corrected AP 30-minute NEE data was reported as 0.20 µmol m−2 s−1 or 6.7% of the daily means (Nicolini, 2012). The Ankasa Resource Reserve, established in 1934 (Hall & Swaine, 1981), lies within the administrative rule of the Jomoro district in the Western region of Ghana and is under the paramount chief of Beyin (Bandoh, 2010). The reserve was managed as a protected timber producing area until 1976 at which time it was designated as the Ankasa Resource Reserve (Damnyag et al., 2013). The forest area is comprised of ∼500 km2 surrounded by deforested landscapes; the area is ∼90 m above sea level with mean annual temperature of ∼25 °C. According to Hawthorne & Abu-Juam (1995), the Ankasa Resource Reserve has an average Genetic Heat Index (GHI) of 301, compared to a maximum of 406 (Janra & Aadrean, 2018; Vanclay, 1998), designating the reserve as a global priority conservation area that should be permanently removed from timber production. Hilly portions of the reserve showed the highest GHI score of 406 (Hawthorne & Abu-Juam, 1995). The high GHI scores in Ghana are amongst the “hottest” patches of genetic rarity in Africa, many of the species concerned being found elsewhere only across the border in Southern La Cote D’Ivoire. Official records on timber logging activities in the Ankasa Resource Reserve are incomplete as the management objective has been primarily for protection and resource conservation, however, illegal logging in the reserve may have occurred during the period of observation. Poor soils of the area have generally discouraged commodities (i.e., cocoa) production and food farmers. The population around the reserve has been historically low for a forest area (Hall & Swaine, 1981), but has experienced dramatic population increase. For example, in 1960, the estimated population was 45,162 but declined to 37,685 by 1970. The results of Ghana’s 1984 population census recorded a jump in population of 70,881, an increase of 88%. According to the 2000 population census, the population of the Jomoro district had increased significantly to 111,348 an increase of 57% since 1984 (Bandoh, 2010). In the recent 2010 census, the population recorded for the district was 150,107 (Bandoh, 2010) representing an increase of 34%. The red color in Fig. 3A, denoting deforestation over approximately the last decade, illustrates the anthropogenic pressure the reserve faces in the future.

Figure 3 Locations of the eddy covariance sites analyzed in this study.

(A) Location of Ankasa Park Tower, Ghana, Africa, (B) Location of Harvard Forest Tower, Massachusetts, USA (Image credits: (A) http://earthenginepartners.appspot.com/science-2013-global-forest, powered by Google Earth (c) 2020; inset map: https://commons.wikimedia.org/wiki/File:Ghana_(orthographic_projection).svg, License: CC BY SA 3.0; (B) Map credit: Google, Delta SQ, NOAA, US Navy, GA, CEBCO Landsat/Copernicus US Geological Survey 2019); inset map: https://commons.wikimedia.org/wiki/File:Massachusetts_in_United_States_(zoom).svg, License: CC BY SA 3.0).

Harvard Forest, Petersham, MA, USA (Fig. 3B)

The Harvard Forest (HF) Environmental Measurement Site tower (42.537755°N, 72.171478°W; US-Ha1) is a component of the Harvard Forest Long Term Ecological Research (LTER) site in Petersham, Massachusetts (Harvard Forest Long Term Ecological Research Site, 2019) and a core site in the AmeriFlux network (US-Ha1). The elevation of the research area of ∼16.18 km2 ranges from 320 to 380 m above sea level (Fig. 1A). NEE data used in this analysis are available online from the Fluxnet2015 database (https://fluxnet.fluxdata.org/doi/FLUXNET2015/US-Ha1). Additionally, current data are available from AmeriFlux data repository (http://dx.doi.org/10.17190/AMF/1246059) and LTER data archives (Environmental Data Initiative; https://doi.org/10.6073/pasta/74fe96d1571db7f15bf6f1c4f53c0c02). The HF tower measurements, initiated in Oct 1991 with closed path CO2 analyzers, provide the longest continuous set of flux measurements in the US (Barford et al., 2001; Urbanski et al., 2007). The mixed deciduous forest stand surrounding the tower has been regenerating on abandoned agricultural land since the late 1890’s punctuated by a major hurricane disturbance in 1938. The Harvard Forest, US-Ha1 data are NEE quality-checked (e.g., Pastorello et al., 2014), gap-filled, filtered and corrected for low friction velocity periods (e.g., friction velocity (u*) less 0.4 m/s) that underestimate nighttime respiration flux (e.g., Urbanski 1996). Uncertainty for US-Ha1 gap-filled data for the years 1992–2004 was reported as ±0.03 tCO2 yr−1(95% Confidence Interval) relative to a mean NEE for the period of −2.42 tCO2 ha−1 yr−1, determined by random sampling of NEE error populations (Richardson et al., 2006; Urbanski et al., 2007). US-Ha1 data are corrected for horizontal advection of CO2 reported of up to 35% of the CO2 budget during summer intervals from 1999 to 2002 (Staebler & Fitzjarrald, 2004) and up to 40% loss of daytime CO2 flux depending on wind speed (Sakai, Fitzjarrald & Moore, 2001). Corrections are routinely applied to account for incomplete flux emphasizing the importance of understanding site conditions at each location to assess optimal and representative flux results. The dataset was read in line by line and processed using Python Libraries (Pandas, NumPy) with txt format. Year, month, day, hour and NEE data were selected for this study. NEE was determined by calculating the mean of 48 half-hour data for each day as µmol CO2 m−2 s−1, converting the value to gC m−2d−1, and summing daily NEE to calculate annual NEE for each year. The US-Ha1 NEE data over the period of record (e.g., 2000–2019) has been interpreted in the context of historical land use including agriculture, reforestation and hurricane disturbance, demonstrating the impact of weather, climate and human activity on NEE (Cogbill, 2000; Compton & Boone, 2000; Barford et al., 2001; Bellemare, Motzkin & Foster, 2002; Urbanski et al., 2007). The Harvard Forest project area has been studied using a variety of remote sensing data to assess species composition across the project area and to understand short and long term responses to climate change (Kim et al., 2018). Hyperspectral and lidar data (https://glihtdata.gsfc.nasa.gov) (e.g., (Kampe, 2010), Airborne Visible InfraRed Imaging Spectrometer (AVIRIS) imagery (https://aviris.jpl.nasa.gov/dataportal/), and Landsat data (Melaas et al. 2016) are also available for the US-Ha1 project area.

Table 1 Comparison of features and benefits for existing forest carbon protocols and the DMFCP.

Protocol features such as inclusion/exclusion of soil respiration CO2 flux, measurement time intervals and compatibility with expanded forest carbon flux measurements of CH4 and N2O are compared for existing forest carbon protocols and the DMFCP. Each feature is discussed in the text.

	Protocol feature	Existing protocols a	DMFCP	Benefit to Landowner	
1	Project tradable units	Total tCO2 equivalent (tCO2eq) is reported for project area (e.g., CO2eq acre-1) without reference to project area	Tons CO2 equivalent (tCO2eq), gC m−2yr −1 (derived from 10 Hz CO2 data, daily, weekly, monthly and annual sums)	Project reporting on an area basis is a fundamental metric of forest carbon sequestration area, (e.g., per acre), a metric not reported in estimation-based protocols	
2	CO2 observations by direct measurement	No	Yes; infrared and laser-based gas analyzer methods for CO2 (10 Hz)	Direct measurement of GHG’s reduces risk of invalidation, increases quality of forest carbon offsets and offers management information	
3	Monitoring implementation	A timber cruise is completed followed by forward and/or backward tree growth model simulations across arbitrary time intervals	A network of observation platforms is established across the project area, the System of Systems (SoS), with diverse sensors including high precision gas analyzers for CO2, N2O, CH4 and micrometerological data comprising the Global Monitoring Platform (GMP); the SoS automates data and reporting for GMP network nodes, including analytical uncertainty	The SoS and GMP commercial products are standardized and designed to deploy as turn-key engineered operations in the field, lowering the cost of NEE measurements and improving the coverage of NEE across large landscapes; estimation protocols are not standardized and do not directly measure CO2 and other gases of interest	
4	Calculation of net forest carbon sequestration	Plot based timber inventory conducted every 6 or 12 years employing non-standardized tree growth simulation models	Use standardized scientifically accepted equation for NEE (NEE = GPP + Reco (Reco = Ra + Rh)), where GPP = gross primary production or photosynthetic assimilation, Reco = ecosystem respiration, Ra = autotrophic respiration by plants, and Rh = heterotrophic respiration by soil microbes. (see Methods section, equations (1) and (2) in the text)	Landowners and stakeholders can rely upon an accepted universal approach to quantify net forest carbon sequestration (NEE) in contrast to estimation protocols that do not employ actual CO2 measurement at any time during the protocol process	
5	Vertical gross and net flux observations	No	Yes; eddy covariance methods are applied resulting in 30” averages of gross vertical CO2 fluxes used to calculate daily/annual net carbon flux (NEE)	Direct measurement reduces risk of invalidation, increases quality of forest carbon offsets and offers management information	
6	Universal metric for Annual Net ecosystem exchange (NEE) or net forest carbon sequestration	No, forest carbon sequestration is based on regionally applied estimation algorithms and growth models as proxies for annual changes in net forest carbon	Yes; vertical CO2 fluxes are used to calculate daily, seasonal and annual NEE reported as ppm CO2 m−2 time−1	Net changes in annual forest carbon sequestration are based on 30” interval data providing daily, weekly, monthly and annually resolved changes in NEE; in contrast estimation-based model runs start and end subject to user discretion and without validation by direct measurement of CO2	
7	Soil CO2 flux	No	Integrated in vertical fluxes	Complete accounting of carbon flux is required for NEE; the DMFCP provides data on soil ecosystem dynamics	
8	Cost to Landowner	Substantial fees are incurred from inception to registry listing of carbon credits; fees increase with size of project	No direct fees from inception to listing on a registry; upfront payments and annual royalty payments may be structured within a project agreement and contract between landowner and service provider	Elimination of direct fees to initiate a forest carbon project incentivizes landowners to engage in forest carbon programs with economic, ecological and business advantages	
9	Time interval to achieve positive revenue	Years (1–5)	Daily to yearly, subject to project agreement and contract	Revenue to landowner is achievable, in practice, based on daily NEE but more typically would be paid annually, or over multiple years resulting in long term incentivizes for sustainable management; traditional protocols may require years to receive initial payment	
10	Marketing and sales of GHG offsets	Responsibility of landowner (e.g., fee-based listing on a registry); voluntary and compliance offsets are priced differently based on discretionary criteria	Projects and products are pooled into portfolios and listed in a no-fee registry for sale to voluntary and compliance buyers worldwide, subject to project agreement and contract	Relieves landowner from handling carbon offsets once issued and from additional cost; direct measurement creates equivalent voluntary and compliance offsets	
11	Baseline	Estimated and uncertain; based on counterfactual arguments and proxy data; positive values are not permitted, or default value is used	Baseline is the zero-emissions point from which positive, negative or neutral emissions of CO2 flux occur; the zero baseline is shared across analyzers via calibration with shared standards and references	All NEE results are instrumentally and financially comparable providing improved management of multiple project landscapes; a measured zero baseline eliminates estimation invalidation risk	
12	Additionality	Based on uncertain counterfactual arguments regarding unobserved CO2 or default values and other criteria	Simple mass balance of carbon (e.g., NEE) across designated areas can be summed to determine overall carbon balance, or test for differences between pooled portfolios offering measured and numerical tests of additionality management plans or contracts	Eliminates uncertainty associated with this factor; provides near real time data for NEE and forest project management planning across additional landscapes and property ownership (e.g., municipal, private)	
13	Invalidation period and compliance testing	Up to 8 years based on 5% invalidation rule	No invalidation period is required as validation with shared universal standards is conducted every 30”; invalidation can be triggered at any time instrument performance is reported as faulty	Elimination of an invalidation period with a near-real time system check will attract more project participants and buyers of carbon project products	
14	Third party verification	Third party validates calculations and estimation protocols reported in project documents; it does not include validation by independent direct measurement	Third party validation is made by independent direct measurements by an unaffiliated group as specified in the governing project agreement and contract	True independent third-party validation will support pricing of GHG products and market transactions as well as provide strict testing for invalid and fraudulent claims of GHG reductions based on direct measurement records	
15	Test for switch to net positive emissions	No	Yes; NEE identifies transitional net negative (i.e., carbon offset producing) to net positive forest carbon dynamics	Switch to positive emissions may suggest landowner management practices to attain net neutral or net negative balance and may indicate changes in forest ecosystem function due to climate change	
16	Permanence	100-year requirement	Up to 100 years but achievable in decadal increments; a 100-year time horizon is an arbitrary project interval	A 100-year forest carbon permanence requirement is a primary barrier to landowner and investor participation; 10-year interval project planning allows extensions or exit and is compatible with short term financial forecasting	
17	Project exit or termination	High penalty	Ton-year accounting is employed to adjust exit penalty based on project impact of atmospheric emissions sequestered over time	Barriers to forest carbon management are lowered when reasonable exit strategies are available based on an accepted accounting method	
18	Monitoring of CH4, N2O and other gases	Not applicable as estimation protocols based on forest mensuration are not intrinsically linked to CH4 and N2O emissions	Eddy covariance can be used to determine next flux for CH4, N2O and other gases employing commercially available instrumentation, similar to the method used for CO2; eddy covariance provides a combined three-gas GHG budget	A three-gas GHG budget offers landowners more options to manage GHG neutral budgets and will expand areas of project applications and increase product options	
19	Incorporate isotopologues of CO2 and other GHG’s	Not applicable as estimation protocols based on forest mensuration are not intrinsically linked to GHG isotopologues	Eddy covariance can be used to determine isofluxes for any isotopologue, similar to the method used for CO2 creating new product categories	Isotopologues of CO2, CH4 and N2O, among others, may offer the landowner additional options to manage projects for net or negative GHG impacts and will increase the diversity of forest carbon product options	
20	Wetland, aquatic and oceanic emissions	Not applicable as estimation protocols based on forest mensuration are not intrinsically linked aquatic/oceanic sources for CO2 and CH4	Eddy covariance can be applied to wetlands, aquatic and estuarine features and to oceanic systems by measurement of CO2 and CH2 exchange with the water surface	Landowners with wetland and aquatic features will benefit from inclusion of these aquatic sources as associated forest GHG products; all stakeholders benefit from expanded knowledge of Earth system function including oceanic GHG dynamics	
21	Contribution to forestry & atmospheric science and climate change studies and models	Lack of publications employing estimation-based protocol data with numerical climate change studies and models	All GHG flux data are relevant to evolving ecosystem function relative to climate change and human activity; all data may be incorporated in climate change research and atmospheric transport models based on calibrated and standardized measurement protocols	All stakeholders benefit from understanding the mechanisms affecting forest GHG dynamics and policies; landowners may employ model data to develop climate change mitigation strategies	
22	Uncertainty and errors	Uncertainty and error sources are estimated at up to ∼82%, representing the contribution of ecosystem respiration to NEE, a quantity not measured with estimation protocols; errors for N2O and CH4 are unknown as forest growth models are not parameterized for these gases and direct measurements are not made	Uncertainties and errors for eddy covariance methodology and calculation of up to ∼20% annual NEE have been reported, and include instrumentation, set up, data processing, and up-scaling NEE from a single tower, to yield 30 min CO2 flux averages; corrective measures are typically applied to sources of uncertainty resulting in errors of ∼±0.03 tCO2 yr−1	Up-scaling from eddy covariance tower data can be addressed by increasing the number of observation platforms and tower heights within the SoS sensor architecture; widely accepted corrections for NEE uncertainties and errors can be uniformly applied to NEE data across the SoS harmonizing uncertainty analysis and corrections that are under active and evolving study in contrast to estimation protocols that, to the best of our knowledge, have not undertaken comparison with directly measured CO2	
23	Underlying Financial Terms and Contract	Estimation Protocols employ typical contract terms but do not include standardized performance metrics based on direct measurement of GHG’s	Project terms and contract language will be standardized including performance metrics, pricing metrics and exit strategies (e.g., item 17) including force majeure clauses and technology performance specifications (Figs. 1 and 2 ; Supplemental Information 1)	Standardized measurement performance terms and contracts apply to all projects, voluntary and compliance, harmonizing efforts for forest conservation and reforestation	
Notes.

a (California Air Resources Board, 2011; California Air Resources Board, 2014; Anadiotis et al., 2019; Kollmuss & Fussler, 2015; Winrock International, 2016; Marland et al., 2017; Climate Action Reserve, 2018; Zhang et al., 2018; Marino, Mincheva & Doucett, 2019)

DMFCP technical description

The DMFCP is comprised of hardware and software components designed to function as an automated commercial field sensor network (Figs. 1 and 2), the System of Systems. (Marino, 2011; Marino, 2012; Marino, 2016c; Marino, 2016d; Marino, 2017a; Marino, 2017b; Marino, 2017c; Marino, 2018a; Marino, 2018b; Marino, 2018c; Marino, 2019; Marino, 2013; Marino, 2014a; Marino, 2014b; Marino, 2014c; Marino, 2015a; Marino, 2015b; Marino, 2016a; Marino, 2016b). An integrated sensor platform, the Global Monitoring Platform, is positioned at each node of the network. The Software components of the SoS are configured to interact with all nodes for automated reporting of data and instantaneous third-party verification of systems, processes, uncertainties and results. The SoS summarizes measurements of GHG fluxes against local, regional and global reference materials for bulk and isotopic composition, providing the basis for calculation of verified tradeable GHG financial products for forests and anthropogenic net carbon fluxes for fossil fuel derived CO2. The DMFCP provides the operational framework for underlying contract terms defining project time periods, land area, management objectives, measurements and cases for intentional and unintentional forest carbon reversals, conditions beyond the scope of this study. Additional details for the SoS, GMP and related field equipment for NEE flux determinations, in addition to typical project agreement and contract terms, are described in Supplemental Information 1. A comparative summary of the features and benefits of the DMFCP and widely employed estimation protocols (e.g., CARB, CAR, ACR, VERRA, CDM) is presented in Table 1.

Figure 4 Net ecosystem exchange for the Harvard and Ankasa times series.

(A) Annual NEE observed at the Harvard Forest, Petersham, MA, USA and at Ankasa Park, Ghana, Africa. (B) Cumulative NEE records corresponding to annual NEE and extrapolated across 100,000 acres are employed for illustration of pre-tax cash flows.

Results

Figure 4 illustrates the annual (tCO2eq) NEE for HF (24 years) and AP (4 years) sites relative to a zero-reference baseline established by instruments (i.e., open or closed path CO2 analyzers) and standard calibration protocols at both sites and to a zero-emissions baseline defining negative (e.g., net CO2 sequestered), positive (e.g., net CO2 emissions released to the atmosphere) or neutral carbon balance (e.g., 0 sequestration/emissions). Annual NEE values for HF and AP were negative over the intervals shown resulting from active forest carbon sequestration and generation of carbon credits (Fig. 4A). Annual NEE for HF ranged from a minimum of −0.53 (2010) to a maximum of −9.09 (2008) tCO2. The mean and standard deviation (SD) for the HF site for 24 years was −4.5 tCO2 acre-1 yr-1 ±2.3 (SD). Annual NEE for AP ranged from a minimum of −6.74 (2013) to a maximum of −15.2 (2011) tCO2. The mean and standard deviation (SD) for the AP site for 4 years was −10.2 tCO2 acre-1 yr-1 ±3.6 (SD). Pre-tax revenue annual variance and risk are illustrated in the HF 2010 NEE (Fig. 2A), emphasizing a reversal of +4.79 tCO2eq relative to 2009, equivalent to a one-year loss of $4,790,00 ($10 tCO2eq), but again reversed the following two years attaining −5.04 tCO2eq and revenue of $4,510,000 ($10 tCO2eq). Figure 4B shows the corresponding cumulative NEE across the observational periods recorded for each site extrapolated to 100,000 acres (40,469 hectares). The HF and AP linear cumulative NEE provides insight into the potential short and long-term sequestration capacity of the respective forest landscapes. The AP NEE slope of −8.40×  is 1.7 times that of the HF suggesting that in this case, the tropical wet evergreen forest site experienced consistently greater sequestration of carbon than the temperate deciduous forest. However, caveats apply in that tropical forests may not result in larger long-term carbon sinks, nor is continued net negative carbon sequestration guaranteed or required for forest carbon trading markets. For example, tropical forests typically have larger gross production but a corresponding larger respiration (Baldocchi, Chu & Reichstein, 2018; Baldocchi & Penuelas, 2019). Additionally, the two forest locations differ in stand age and history of disturbance, factors that are known to affect NEE (Hollinger et al., 2013; Ouimette et al., 2018; Urbanski et al., 2007). However, NEE provides a quantitative record of daily and annual sums of carbon sequestration characterizing the fundamental nature of derivative carbon products that cannot be replicated by proxies for forest carbon sequestration (e.g., estimation-based protocols). Annual NEE trends may also be difficult to characterize for sites with less than five years of NEE data emphasizing the importance of establishing new and sustained NEE observation platforms (Baldocchi, 2019); Dennis (Baldocchi, Chu & Reichstein, 2018). Figure 5 illustrates landowner pre-tax cash flow (millions USD) relative to variable carbon pricing of tCO2eq ($5, $10, $15, $36) for cumulative NEE consisting of 24 and 4 years for the HF and AP sites, respectively. The values represent extrapolations of measured local NEE to 100,000 acres (40,469 hectares) multiplied by the annual NEE record for each site. Two cases are represented in which the landowner receives a single upfront payment (Case 1) or an upfront payment plus annual royalty on sales (Case 2). Case 1 pre-tax cash flow estimates range from upfront payments (e.g., 10%) of $230,000 to $1,670,00 and $510,000 to $3,680,000 for HF and AP, respectively, across carbon prices of $5 to $36 tCO2eq. Case 2 pre-tax cash flow estimates range from an upfront payment (e.g., 8%) plus deferred payouts based on realized revenue from the sale of all carbon products (e.g., 6%) of $3,520,000 to $25,360,000 and $1,640,000 to $11,790,000 for HF and AP, respectively, across carbon prices of $5 to $36 tCO2eq. Variance for the total pre-tax sales value of ±20% of realized revenues is indicated by vertical bars to reflect uncertainty in the sale of carbon products for Case 2.

Figure 5 The pre-tax cash flow for two hypothetical cases for landowner revenue associated with forest carbon management.

The graph depicts projected cash flows for landowners for the two cases described for Harvard Forest, USA, and Ankasa Park, Africa. Upfront payments are paid to the landowner prior to project initiation. Additional cash flows are created by selling carbon products after the initial year of monitoring (Fig. 1). Case 1 (unfilled bar, Harvard Forest; filled black bar, Ankasa) shows the total pre-tax cash flow for an upfront payment of 10% of the projected annual revenue. Case 2 (light shaded bar, Harvard Forest; dark shaded bar, Ankasa) shows the total pre-tax cash flow for an upfront payment of 8% of the projected annual revenue plus deferred payouts of 6% of the realized revenue from the sale of all carbon products. The vertical bars represent the impact of a ±20% market variance on realized revenue. These examples are provided for purposes of illustration and do not represent actual carbon products by type or cashflow.

Figure 6 illustrates cases of pre-tax cash flow change for a decrease/increase in native carbon sequestration strength based on the minimum, mean and maximum values of NEE observed for each site’s historical record (extrapolated to 100,000 acres or 40,469 hectares). Local sequestration strength is expected to vary annually in response to rainfall and related ecological factors. We use the minimum, mean and maximum values for NEE recorded at each site to illustrate the effect of variable annual sequestration rate on pre-tax revenue. Project value ranges from $760,000 to $13,830,000 and from $2,140,000 to $4,860,000 across the minimum, mean and maximum values for the annual records of the HF and AP sites, respectively.

Figure 6 Projected pre-tax cash flows for the Harvard and Ankasa forest over the time series studied.

Landowner pre-tax cash flows are depicted based on a price of 10 per tCO2eq across the minimum, mean and maximum values recorded for the Harvard Forest, USA (unfilled bars), and the Ankasa Park forest, Africa (filled bars), extrapolated to 100,000 acres for the historical record of each site. The vertical bars represent the impact of a ±20% market variance on realized pre-tax revenue. These examples are provided for purposes of illustration and do not represent actual carbon products by type or cashflow.

Figure 7 illustrates pre-tax cash flows for mixed carbon product types and pricing for Case 2; example product inventory and pricing for the products is indicated below each set of bars. Note that the hypothetical carbon products range in price from $12 tCO2eq for compliance offsets to $50 tCO2eq for carbon products with the additional element of biodiversity (e.g., Genetic Heat Index and conservation of floral and faunal species). Total pre-tax cash flow for Case 2 is $16,380,000 and $7,610,000 for the HF and AP sites, respectively. These data illustrate the higher potential revenue based on sale of mixed products and pricing for voluntary, compliance and regulatory markets. The vertical bars for Case 2 represent 20% variance in market uncertainty.

Figure 7 Hypothetical mixed carbon product types and projected pre-tax cash flows based on the example product inventory noted.

Total pre-tax cash flow for the Harvard Forest, USA (light shaded bar), and the Ankasa Park, Africa (dark shaded bar), is $16,380,000 and $7,610,000, respectively. Both project projections illustrate the potential value of offering a mix of products and pricing to maximize revenue. Products may also incorporate additional GHG’s (e.g., CH4, N2O), isotopic species of the GHG’s, aspects of the project land and cultural features related to landownership and stewardship. These examples are provided for purposes of illustration and do not represent actual carbon products by type or price.

Discussion

The SoS and DMFCP features continuous eddy covariance measurements for determination of NEE for forest carbon providing standardized commercial methods and operations (Figs. 1 and 2) in contrast to estimation based protocols that do not observe CO2 assimilation via photosynthesis or efflux via respiration. Shared calibration of instruments and reliance on a shared zero-emissions flux baseline (e.g., carbon negative, neutral or positive) ensures that all analyzers and results (e.g., SoS and GMP sensor nodes) within a network or between networks (e.g., SoS) are comparable, inclusive of analytical uncertainties (Table 1). The near real-time data (i.e., 30-minute average of 10 Hz CO2 measurements) for forest NEE achievable with the eddy covariance sensor of the DMFCP offers insights into forest carbon dynamics and ecosystem function previously unavailable to landowners, investors and related stakeholders (Baldocchi, 2019). The result is a first of its kind pooled portfolio of diverse forest projects and harmonized products for sale to voluntary and compliance buyers worldwide transacted as tCO2eq (Figs. 1 and 2). The DMFCP incentivizes forest conservation efforts, communities and management of atmospheric CO2 emissions compared to estimation-based protocols (Table 1) and REDD + platforms that rely on such protocols (Köhl, Neupane & Mundhenk, 2020). NEE uncertainties can be quantified and corrected for each project (e.g., single, multiple networks) according to established and evolving methods within the forest carbon research community (e.g., Vitale, Bilancia and Papale, 2019b), particularly in conjunction with remotely sensed data. Commercialization of established forest carbon research methodologies is feasible and applicable to forest projects worldwide.

The NEE sites described in this work representing tropical and deciduous forests, when pooled as a portfolio, provide species and ecological diversification with respect to NEE source strength, vulnerability to climate change, population pressure and external risks (e.g., currency value, national/sub-national environmental regulation) (Tarnoczi, 2017), a common investment risk reduction approach employed in modern portfolio theory (Busby, Binkley & Chudy, 2020; Paut, Sabatier & Tchamitchian, 2020). For example, while HF experienced the lowest NEE during 2010 (−0.59 tCO2 acre−1 yr−1, −0.4 tC ha−1yr−1), a period known to be associated with anomalous drought and up to 86% respiration relative to gross primary productivity (Gonsamo et al., 2015; Munger, Whitby & Wofsy, 2018), AP experienced the highest NEE of the available record (−15.2 tCO2 acre−1 yr−1, 10.2 tC ha−1yr−1), in part offsetting the HF loss. A portfolio with diverse projects would be similarly buffered from extreme changes. Landowner agreements and contracts could also specify options for aggregation of annual data intervals to buffer extreme weather conditions (excluding catastrophic events such as wildfire, multiyear drought, and hurricane) (see Supplemental Information 1). The upfront and royalty revenue structure resulting from sale of DMFCP products, proposed in this study, provide financial incentive for the landowner to rapidly enter into reforestation and forest management projects in lieu of deforestation (e.g., legal and illegal) and increased anthropogenic disturbance. Given the high rates of population growth within the AP reserve area (∼298% population increase from 1970 to 2010), revenue programs may be uniquely suited for preservation and management of protected areas in conjunction with community based efforts (e.g., Bempah, Dakwa & Monney, 2019). Long-term forest carbon projects are likely to increase harvest ages and management of forest stocking for optimal forest growth while promoting carbon benefits of active sustainable forestry (Bastin, 2019; Chazdon & Brancalion, 2019); Chazdon, 2008). Enhancement of biodiversity, food webs and cultural engagement may also accrue as forests grow (Bremer et al., 2019; Li et al., 2019; Watson et al., 2018). Conservation and commercial forestry operations, although likely to have different goals, are accommodated by the features and benefits of the DMFCP for effective carbon management.

The hypothetical financial structure and cases for pre-tax revenue for landowners illustrate the potential impact of the DMFCP. The long-term cumulative value of both sites, shown in Figs. 4B and 7 (e.g., Total revenue from mixed products and pricing), benefit landowner property valuation and reduces cost of delayed reforestation in-line with indices for value of timber land operations (Ferguson, 2018; Keith et al., 2019; Zhang, 2019). Figure 7 emphasizes the revenue potential of mixed forest carbon products incorporating features of project biodiversity, such as noted for AP by high Genetic Heat Indices of up to 401 (Hawthorne & Abu-Juam, 1995), and allocation of offsets for specific markets. Pre-tax revenue for mixed carbon products and pricing is projected at up to $16,380,000 for the HF over the 24-year period (Fig. 6), an ∼5×  and ∼2× return compared to pricing of $5 and $10 tCO2eq (Fig. 5, Case 2), respectively, covering voluntary and compliance carbon pricing levels (Hamrick & Gallant, 2018; World Bank Group, 2015). The two sites, irrespective of the differing time-series length, actively sequester carbon at different rates; it is not known if the observed trends will reverse as a result of climate change and/or anthropogenic activity. The requirement for long term CO2 measurement cannot be understated for determination of variance in annual changes of NEE and for creation of corresponding annual forest carbon financial products resulting from NEE (Baldocchi & Penuelas, 2019; Marino, Mincheva & Doucett, 2019; Munger, Whitby & Wofsy, 2018). For present purposes we assume that 100% of the products are sold in each case covering the cost of the DMFCP.

There is no single figure of merit for NEE uncertainty. One of the main concerns with eddy covariance based NEE, applicable to establishing networks of eddy covariance towers as proposed, is the upscaling of limited footprints for individual EC towers to surrounding ecosystems (Table 1, #22) Baldocchi, 2003; Kumar et al., 2016; Ran et al., 2016; Román et al., 2009). Up-scaling is particularly important for mixed forest projects wherein changing wind direction alters the source weight of heterogeneous land cover (Kim et al., 2018) and remains a challenge to large-scale NEE determinations including use of very tall towers (e.g., >300 m) and mesonet configurations to expand eddy covariance footprints (Andrews et al., 2014; Chi et al., 2019; Goulden et al., 2006). SoS architecture and sensor placement details will vary for each project addressing sources of uncertainty established by initial survey and temporary placement of SoS platforms for evaluation. Scale-up of eddy covariance flux tower data combined with remote sensing data is under active study and directly relevant to developing approaches for the SoS (Fang et al., 2020; Kenea et al., 2020; Peltola et al., 2019; Xiao et al., 2008). In actual DMFCP implementation for HF and AP sites reported here (Fig. 3), remote sensing data would be used to establish reliability of data extrapolation from a single tower and to guide the placement of additional towers to fill spatial gaps in NEE measurement. A minimum two-tower configuration or paired GMP sensor nodes for larger networks, would provide redundancy and cross-checks to report SoS NEE combined uncertainty (e.g., He et al., 2010; Post et al., 2015; Griebel et al., 2020). In addition, the use of open source eddy covariance processing software such as ONEFlux (https://ameriflux.lbl.gov/data/download-data-oneflux-beta/; see Supplemental Information 1, Eddy Covariance) and commercial software (e.g., https://www.licor.com/env/products/eddy_covariance/software.html), applied uniformly across the SoS would harmonize data treatment including uncertainties for CO2 and CH4 flux (e.g., Richardson et al., 2019).

Accepting NEE uncertainties (e.g., ‘Methods’ section), we argue that the approach is a game-changer for creation and verification of forest carbon financial products compared to estimation and model simulation-based protocols. For example, terms defined in (1) and (2) (Methods section) are not defined or measured in estimation protocols (e.g., CARB, CAR, CDM, ACR, VERRA), unavoidably introducing fundamental uncertainties in NEE rendering the basis for reporting gC m−2 yr−1 for NEE as problematic and unverifiable (Marino, Mincheva & Doucett, 2019). Moreover, interpreting estimation based timber inventory protocols as representing only above ground carbon (e.g., photosynthetic assimilation) likely results in over crediting errors given that ecosystem respiration accounts for up to ∼82% of gross carbon flux from the soil to the atmosphere (Baldocchi & Penuelas, 2019; Bond-Lamberty et al., 2018; Giasson et al., 2013; Richardson et al., 2013). The anomalously low NEE for HF year 2010 (−0.53 tCO2eq), associated with drought demonstrates the requirement for ecosystem respiration measurement for NEE. NEE establishes 30-minute flux data comprising detailed baseline resolved time-series for each project yielding annual mean data based on 17,520 such intervals for each CO2 analyzer. To our knowledge, estimation protocols have not been directly compared with CO2 measurements, or peer reviewed, (e.g., California Air Resources Board, 2011; California Air Resources Board, 2014; California Air Resources Board, 2015a; California Air Resources Board, 2015b) limiting scientific acceptance and demonstrating a need for improved and peer reviewed non-NEE based methods.

In addition to potential revenue for landowners, the DMFCP simplifies the forest carbon protocol process compared to traditional approaches that differ in methods, assumptions and allowance for discretionary revisions (Kollmuss & Fussler, 2015; Marino, Mincheva & Doucett, 2019). A summary of DMFCP protocol features and benefits to landowners is provided in Table 1, with reference to Figs. 1 and 2. Equivalent units of tCO2eq or units as converted are employed for the DMFCP and traditional protocols (#1), noting (e.g., tCO2eq acre-1) that estimation protocols do not report carbon sequestration according to project area, potentially misleading landowners. Items #2 - 7 have been covered above, defining the insuperable differences between direct measurement of CO2 versus the use of proxies (i.e., CO2 is not directly observed at any time in the estimation protocol process) that do not provide data for quantifying NEE (e.g., ecosystem photosynthesis and respiration) according to accepted universal scientific practice (i.e., Item 4, Table 1, Eqs. (1) and (2), Methods section).

Revenue and time-to-revenue are key factors in landowner forest carbon project participation. Traditional protocols (e.g., CARB, CAR, ACR, VERRA, CDM) require lengthy periods (e.g., 2–5 years) of fee-based project certification and registration prior to payment, limiting landowner participation (Kerchner & Keeton, 2015; Köhl, Neupane & Mundhenk, 2020). In contrast, the DMFCP process can provide an upfront payment and annualized payment (e.g., case 2, Figs. 5 and 7) in a no-fee agreement (Fig. 2) available immediately via cell phone payment according to a governing agreement (e.g., contract) that also includes a no-fee listing in an open source registry (summarized by #8,9,10, Table 1). The DMFCP embodied in the SoS and GMP obviates three features intrinsic to traditional protocols including elimination of baseline estimation (#11), tests for additionality (#12), and a multiyear invalidation period (#13) linked to compliance testing and third-party verification (#14). Direct measurement establishes forest carbon flux as either negative (e.g., CO2 sequestration), positive (e.g., CO2 efflux), or zero (sequestration balances efflux)—measurements cannot be made retrospectively. It follows that a zero-emissions baseline is intrinsic to a time-series of positive/negative/zero NEE measurements (Fig. 2A) integrating forest tree species, vegetation and carbon fluxes across and within the project area including all above and below ground carbon fluxes (DiRocco et al., 2014; Urbanski et al., 2007). DMFCP carbon accounting is not subject to uncertainty related to selection for species distribution and growth simulation models typical of traditional protocols (Kollmuss & Fussler, 2015). Additionality tests require a counterfactual argument (Ruseva et al., 2017) that cannot be validated and is subject to discretionary adjustment. A credit is considered additional if the emissions reduction that underpins the credit would not have occurred in the absence of the activity that generates the credit (Kollmuss & Fussler, 2015). In contrast, the DMFCP results in near-real time (30-minute average of 10 Hz measurements) NEE time series and trends (Dou & Yang, 2018), obviating reliance on uncertain project scenarios and an impractical prediction of future emissions against possible forest disturbance. Further, tests of net emission reduction across project areas or jurisdictions for specified periods of time can be readily calculated from DMFCP results for independent projects, establishing simple numerical additionality (Fig. 2B) rules for established private and public lands, as could be adopted by municipal and private entities. The DMFCP does not require an invalidation period (# 13) compared to estimated forest carbon offsets. In contrast to long inspection intervals for traditional forest carbon protocols (e.g., 6 or 12 years; California Air Resources Board, 2011, 2015), the DMFCP results are subject to instantaneous invalidation by third party inspection and routine flags for anomalous operation within the SoS. The DMFCP is subject to replication of equipment and system performance standards, precision and accuracy of universal references and review of NEE from raw data to financial products at any time. The DMFCP employs a real-time wireless reporting and verification concept of operations architecture including third-party independent observers of all data developed for each SoS network (Anadiotis et al., 2019) with invalidation authority (#13). In contrast, third party validation for CAR projects, for example, is based on desk and paper review of unobserved CO2 (e.g., proxies) and cannot support instantaneous spontaneous invalidation testing and enforcement.

Once a project is in operation, a switch from carbon negative to carbon positive ecosystem function is key to project management, revenue projections, accounting and contract terms and to an understanding of ecosystem function in relation to climate change and anthropogenic activity. Traditional forest carbon protocols do not appear capable of determining when a forest project switches to net positive emissions to the atmosphere on an annual basis; the DMFCP NEE measurements provide this diagnostic (#15). Item #15 is also linked to demonstration of project permanence (#16) and termination of a project (#17). Traditional protocols require an arbitrary 100-year period of monitoring and maintenance for project carbon with a punitive penalty for early termination; lack of CO2 measurement renders both factors indeterminate, impractical and biased against the landowner. The DMFCP employs ton-year accounting, an IPCC recognized method that does not impose an artificial time horizon for tree growth (e.g., 100 years) opening forest carbon sequestration projects to a wider range of forest project types and project intervals (Cunha-e-Sá, Rosa & Costa-Duarte, 2013; Levasseur et al., 2012). The ton-year accounting method accommodates combined budgets of CO2, CH4 and N2O resulting in a comprehensive and realistic net GHG project budget (Courtois et al., 2019; Richardson et al., 2019), an approach that can be applied to the spectrum of projects from pure conservation to working forests, however, not achievable with estimation-based forest carbon denominated protocols.

Items 1 to 17 for existing protocols address two key factors favoring deforestation engagement: transaction requirements and liquidity. Forestland as a timber asset requires long periods of growth to harvest and is generally financially illiquid until harvested (Mei, 2015). It is argued here that business development of forest carbon projects, as practiced according to traditional protocols, is overly cumbersome and lengthy to establish offset transactions, and financially inviable to compete with the short time intervals of deforestation often resulting from illicit transactions (Alam et al., 2019; Tacconi et al., 2019; Tellman, 2016). In addition, with the use of satellite imagery, illegal and non-conforming deforestation can be detected in near real-time, with spatial resolution of meters limiting potential gaming of the system and uncertainty in the sources of CO2 flux (Hayek et al., 2018; Tang et al., 2019). Rapid set-up of the SoS direct measurement platform, no-fee based agreements, upfront and annualized payments, discrete revenue intervals of 10 years, and reasonable exit terms align landowner business operations (private and commercial) within realistic financial frameworks to potentially deter rapid deforestation within culturally diverse transactional and transnational frameworks (Fenichel et al., 2018). Additional points of comparison concern the limitation of traditional protocols to accommodate the spectrum of relevant GHG’s (e.g., N2O, CH4, PFC’s, HFC’s, SF6, NF3) (#18), isotopologues of GHG’s (e.g., 13CO2, 14CO2) (#19), the inclusion of aquatic features (e.g., rivers, lakes ponds, wetlands, oceans) (#20) and the lack of contribution to ecosystem science and climate change studies and models (#21). Traditional forest carbon protocols (Kollmuss & Fussler, 2015) were developed for singular application to forests, incorporating methodology employed for timber management and primarily restricted to capturing above ground carbon. As a result, algorithms developed for forest CO2 are not readily applicable to other GHG species and diverse biospheric landscapes. Based on the comparisons, the insuperable shortcomings of traditional protocols do not provide data that contribute to the evolving science of forest carbon sequestration, climate change studies and related model development that are well established in the growing NEE methodology (Baldocchi, 2019). Climate change impacts on forest carbon storage are not included in project risk for estimation based protocols (California Air Resources Board, 2015b) even though soil carbon efflux over the 100-year required period is likely to respond to global warming and changes in precipitation (Amundson & Biardeau, 2018; Bond-Lamberty et al., 2018; Schlesinger & Amundson, 2018).

The DMFCP can be applied to international emission reduction policies recognizing scientifically accepted methods, shared NEE data processing algorithms, elucidation of uncertainties (#22), standardized terms and contracts for voluntary and compliance offsets (#23, Supplemental Information 1) including clauses for reversal of net forest carbon sequestration due to intentional or unavoidable natural conditions (e.g., fire, hurricane, drought). For example, the expansion of measurement networks, data integration and carbon trading are key but unrecognized components of the Paris Agreement (Clemencon, 2016; Rimmer, 2020), and REDD+ programs (Foss, 2018). For example, Article 10 of the Paris Agreement, lacks guidance on how pledged and claimed reductions that are non-binding will be verified and traded (Ollila, 2019; Rimmer, 2020; Spash, 2016), shortcomings that are mitigated by the DMFCP. The estimation approach remains embedded in the United Nations Framework Convention on Climate Change (UNFCCC) (UNFCCC, 2013) that promulgated reporting of emissions based on estimation, rather than direct measurement, an approach constraining advancement of carbon credit trading. According to the UNFCCC approach, estimates of greenhouse gas emissions are inventoried and multiplied by an emission factor to yield a national emission rate for each source and each greenhouse gas (Cheewaphongphan et al., 2019; Van Vuuren et al., 2009). Emissions of Kyoto gases are multiplied by the Global Warming Potential for each gas specifying the radiative efficiency as a warming agent for each gas relative to that of carbon dioxide over a 100-year time horizon (Kollmuss & Fussler, 2015). The resulting estimation for national emission inventories, used by vendors and policy platforms (e.g., REDD+), are widely acknowledged as flawed and inaccurate (Jonas et al., 2019; Pacala et al., 2010). Importantly, the estimation data are not directly comparable across diverse ecosystems lacking shared standards and universal measurement methodology. The DMFCP updates the UNFCCC and REDD+ methods to validate and monetize claims of emission reduction and to determine GHG budgets across diverse ecological landscapes at the national and sub-national levels fulfilling the Paris Agreement (e.g., Article 10) and REDD goals and objectives. The DMFCP normalizes forest emission reduction determinations for voluntary and compliance markets bridging the gap between methods, project types and outcomes for stakeholders.

Conclusions

The DMFCP comprises a commercial standardized measurement-to-monetization system for the determination of NEE. NEE enables creation of verified forest carbon financial products contributing to the improvement of methods that underpin large-scale forest conservation and reforestation, a global problem of high importance in the management of anthropogenic climate change. The SoS and GMP components can be applied to GHG’s across large-scales and diverse locations, corrects traditional carbon credit gaps in validation and recalibrates equivalent voluntary and compliance programs that rely on them such as the CARB, CAR, ACR, VERRA and CDM, as well as the REDD+ and Paris Agreement platforms. The DMFCP, coupled with contributions of the forest carbon research community to commercialization efforts, and updated policies, can address the ∼0.9 billion hectares of restorable landscapes, offering a viable approach to retain the Earth’s natural protective capacity to sequester atmospheric CO2 now and for future generations.

Supplemental Information

Supplemental Information 1 System of Systems Technical Description

The System of Systems (SoS), the Global Monitoring Platform (GMP), the eddy covariance method and the Project Management Plan are described, amplifying sections in the narrative.

Click here for additional data file.

Supplemental Information 2 Raw Data for Annual NEE

The raw data for annual NEE (tCO2eq acre-1 yr-1) is provided for the Harvard Forest (HF) (US-Ha1) and the Ankasa Park (AP) (GH-Ak1) sites analyzed in this study.

Click here for additional data file.

Additional Information and Declarations

Competing Interests

Author Contributions

Patent Disclosures

Data Availability

1 “Carbon dioxide equivalent” or “CO2eq” is a term for describing different greenhouse gases in a common unit. For any quantity and type of greenhouse gas, CO2eq is a term for describing different greenhouse gases in a common unit. For any quantity and type of greenhouse gas, CO2eq signifies the amount of CO2 which would have the equivalent global warming impact.

2 The term isotopologue refers to chemical species that differ only in the isotopic composition of their molecules or ions.

3 The term stable isotope has a similar meaning to stable nuclide but is preferably used when speaking of nuclides of a specific element. The expression “stable isotope ratio” is used to refer to isotopes whose relative abundances are affected by isotope fractionation in nature. The stable isotopic compositions of low-mass (light) elements such as oxygen, hydrogen, carbon, nitrogen, and sulfur are normally reported as “delta” (d) values in parts per thousand (denoted as ‰) enrichments or depletions relative to a standard of known composition. The symbol ‰ is spelled out in several different ways: permil, per mil, per mill, or per mille. The term “per mill” is the ISO term, but is not yet widely used. d values are calculated by: (in ‰) = (Rsample/Rstandard - 1)1000 where “R” is the ratio of the heavy to light isotope in the sample or standard. For the elements sulfur, carbon, nitrogen, and oxygen, the average terrestrial abundance ratio of the heavy to the light isotope ranges from 1:22 (sulfur) to 1:500 (oxygen); the ratio 2H:1H is 1:6410. A positive d value means that the sample contains more of the heavy isotope than the standard; a negative d value means that the sample contains less of the heavy isotope than the standard. A d15N value of +30 ‰ means that there are 30 parts per thousand or 3 ‰ more 15N in the sample relative to the standard.

Bruno D.V. Marino is an Academic Editor for PeerJ. A competing interest is disclosed by Bruno D.V Marino as the author of the patents cited. The data and scientific analyses, while citing the patent literature as supportive of potential real-world applications, was conducted independently of the work presented. Data presentation, analysis, and interpretation are based on established scientific principles and reported in an objective manner. Bruno D.V. Marino and Vinh Truong are unpaid associates of Planetary Emissions Management Inc. Richard Gyimah is a paid employee of the Forestry Commission of Ghana. J William Munger is a paid employee of Harvard University.

Bruno D.V. Marino conceived and designed the experiments, performed the experiments, prepared figures and/or tables, authored or reviewed drafts of the paper, and approved the final draft.

Vinh Truong analyzed the data, prepared figures and/or tables, authored or reviewed drafts of the paper, and approved the final draft.

J. William Munger and Richard Gyimah analyzed the data, authored or reviewed drafts of the paper, and approved the final draft.

The following patent dependencies were disclosed by the authors:

Marino, B. D. V. (2011) System of systems for monitoring greenhouse gas fluxes, European Patent Convention patent #2391881.

Marino, B. D. V. (2012) System of systems for monitoring greenhouse gas fluxes, Hong Kong patent #1165004.

Marino, B. D. V. (2013) System of systems for monitoring greenhouse gas fluxes, United States of America patent #8595020.

Marino, B. D. V. (2014a) System of systems for monitoring greenhouse gas fluxes, Japan patent #5587344.

Marino, B. D. V. (2014b) System of systems for monitoring greenhouse gas fluxes, Mexico patent #319180.

Marino, B. D. V. (2014c) System of systems for monitoring greenhouse gas fluxes, Mexico patent #326190.

Marino, B. D. V. (2015a) System of systems for monitoring greenhouse gas fluxes, Australia patent # 2010207964.

Marino, B. D. V. (2015b) System of systems for monitoring greenhouse gas fluxes, United States of America patent #9152994.

Marino, B. D. V. (2016a) System of systems for monitoring greenhouse gas fluxes, China (People’s Republic) patent #ZL201080015551.5.

Marino, B. D. V. (2016b) System of systems for monitoring greenhouse gas fluxes, Japan patent #5908541.

Marino, B. D. V. (2016c) System of systems for monitoring greenhouse gas fluxes, Korea patent #10-1648731.

Marino, B. D. V. (2016d) System of systems for monitoring greenhouse gas fluxes, United States of America patent #9514493.

Marino, B. D. V. (2017a) System of systems for monitoring greenhouse gas fluxes, Australia patent #2015203649.

Marino, B. D. V. (2017b) System of systems for monitoring greenhouse gas fluxes, China (Peoples Republic) patent #CN102405404.

Marino, B. D. V. (2017c) System of systems for monitoring greenhouse gas fluxes, Republic of Korea patent #10-1699286.

Marino, B. D. V. (2018a) System and methods for managing global warming, Canada patent #2813442.

Marino, B. D. V. (2018b) System of systems for monitoring greenhouse gas fluxes, Canada patent #2751209.

Marino, B. D. V. (2018c) System of systems for monitoring greenhouse gas fluxes, Hong Kong patent #1242029.

Marino, B. D. V. (2019) System of systems for monitoring greenhouse gas fluxes, India patent #311228.

The following information was supplied regarding data availability:

The raw data for NEE and CARB-CAR sites are available in the Supplementary File. The data are also available from FLUXNET2015 (https://fluxnet.fluxdata.org/data/fluxnet2015-dataset) under Tier One data following the guidelines of the CC-BY-4.0 data usage license (Attribution 4.0 International (CC BY 4.0); https://creativecommons.org/licenses/by/4.0/). That license specifies that the data user is free to Share (copy and redistribute the material in any medium or format) and/or Adapt (remix, transform, and build upon the material) for any purpose. https://fluxnet.fluxdata.org/data/data-policy/.

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
