# Peer review of "Direct measurement forest carbon protocol: a commercial system-of-systems to incentivize forest restoration and management"

_PeerJ, doi:10.7717/peerj.8891_

## Round 0.1 · original submission · Minor Revisions

In general, the reviewers give a positive evaluation of your work. Thought, they ask for some minor changes, including a deeper discussion of the limitations of the method applied, which is not - in their opinion - as clearly presented as it should be.

·

Basic reporting

The English, reviews and material is clear. I question whether we really need the
lengthy recitation of patents. This clogs the reference list and, to be honest, seems
difficult for me to understand what is being patented. This is not my area!

Experimental design

Applying field eddy flux measurements to determining the effectiveness of a forest to sequester carbon is a new, clean idea. The methodology is oversold. The limitations of the method are not as clearly presented as they should be.

I wrote my review as a piece, and have put this piece in section 3.

Validity of the findings

Review of:
“Direct Measurement of Forest Carbon Sequestration: A Commercial System-of-Systems to Incentivize Forest Restoration and Management”,
by Marino, Truong, Munger and Gyimah.

Reviewed by: David Fitzjarrald, Atmospheric Sciences Research Center, SUNY Albany
(N.B. I do not submit anonymous reviews.)

General comments.
My experience with direct measurements of carbon flux is that of a micrometeorologist. I’ve been around this work for some decades now. However, I have no background in the economics of financing carbon sequestration. My main reaction in reading this material is that the first author has the inverse experience, and those co-authors who have the technical background have not fully argued the case for understanding a number of limitation to the method.

Most of the issues that come to mind have to do with repeated claims that the eddy flux approach delivers values at half-hourly intervals. That is the way the tables that the authors downloaded do present the data, but to assure that these numbers represent net ecosystem exchange, corrections must be made before the final data set is released. This is what rankles about the statement that the method is (line 247) “high precision, high frequency…” It will be more convincing when the authors outline how much better this approach is than competing ones, whatever they are.

These include: Correction for the chronic underreporting of nocturnal fluxes, which bias the results to excessive overall uptake, an error memorialized for all time by a paper by Grace et al. in 1990’s extolling exaggerated C uptake in the Amazon forest. The correction in force for some years now has been simply to assert that the respiratory emission on windy, well-mixed nights holds for the other nights. Nearly twenty years ago we quantified that 10-20% of the CO2 escapes horizontally at Harvard Forest (Staebler and Fitzjarrald, 2004). There is another bias in filtering to fit into the half-hourly data window which can, on certain kind of days, underestimate daytime fluxes (Sakai et al., 2001). I am not certain how the error limits on the Harvard Forest data are estimated, but I would hope that these effects are included. There is the ‘science’ of gap-filling—making up data when the instruments fail—introduces another problem. Finally, the authors imply that the eddy flux method applies to entire enormous forests, but they are certain to know that the ‘footprint’ for flux measurements is only a small area—at Harvard Forest it has been reported to be 0.23 km2 (Kim, 2015 ). (The species composition at Harvard Forest For depends on wind directions, for example.) Where is the discussion of the degree of homogeneity of the forest area being considered? Much of the discussion of the Ghanaian forest has to do with the species richness and paucity of commercial timber. These are truly important considerations but largely outside the purview of carbon uptake measurements. Perhaps the authors do not take these issues to be ‘show-stoppers’ given their urgency to commercialize the product. However, since critics of their commercial proposal could easily be as adept as am I, the authors should tighten their case. As it stands, much of their text resembles what one might hear from a manufacturer of eddy flux equipment, Dr. Burba, for example. Optimism is only sometimes a virtue.
Another difficulty with this paper is the comparison of four years of data from the tower in Ghana with the longest eddy flux record anywhere, at Harvard Forest. I believe that this paper would be greatly served by a careful stating of the uncertainties in the eddy flux method, a plan for assessing the similarity of the forest being considered with the ‘footprint’ of the eddy flux measurement being made. Most convincingly this would be done with some remotely sensed data.

Please help me to see how the remuneration plan works on years, possibly long stretches of time during which a measured forest loses carbon over time. It appears that C uptake at Harvard Forest can be quite variable. Last year, coauthor Munger et al. (2018) noted:
At Harvard Forest: “Annual NEE ranges from -0.4 to -6.1 Mg-C ha-1y-1 (always carbon uptake), with a mean of -2.9 (±1.5) Mg-C ha-1 y-1 . Here we focus on the highest and lowest carbon uptake years, 2008 and 2010, respectively in order to examine how forest carbon balance can shift so dramatically over a 2 year interval.”
If the system were implemented, would the landowner have to pay someone back if the forest is a net carbon emitter? What happens in the case of a wildfire? (Actually, one could follow the other should there be a prolonged drought.) Please try to address these issues.

Specific comments.

1. Line 141 “Genetic Heat Index” not common knowledge. A reference is needed.
2. Line 181. The absurd number of citations to patents fills up the bibliography. Kindly explain to the naïve reader just what is being patented here. Put this information into Appendix 1.



References.

Kim J., 2015. Carbon and water cycles in mixed-forest catchments: ecohydrological modeling of the influence
of climate variability and invasive insect infestation, PhD diss. Boston University.

Munger, J.W., Whitby, T.G. and Wofsy, S.C., 2018, December. Rapid Shifts in Annual Carbon Balance For a Temperate Deciduous Forest: Validating and Diagnosing Ecosystem Surprises. In AGU Fall Meeting 2018 Abstracts.

Sakai, R.K., Fitzjarrald, D.R. and Moore, K.E., 2001. Importance of low-frequency contributions to eddy fluxes observed over rough surfaces. Journal of applied meteorology, 40(12), pp.2178-2192.

Staebler, R.M. and Fitzjarrald, D.R., 2004. Observing subcanopy CO2 advection. Agricultural and Forest Meteorology, 122(3-4), pp.139-156.

Reviewer 2 ·

Basic reporting

In general, the manuscript has gotten some potential to published in the PeerJ journal.
Nevertheless, some issues need to be considered before accepting the manuscript for publication. The manuscript needs some further improved before to be accepted for publication. In general, there are still some occasional grammar errors through the manuscript especially the article ''the'', ''a'' and ''an'' is missing in many places, please make a spellchecking in addition to these minor issues. The reviewer has listed some specific comments that might be helpful of the authors to enhance the quality of the manuscript further. Please consider the particular comments listed below!

Experimental design

• The abstract is well written. The structure is fine.
• Please remove the abbreviation.
• The method section in general, is well written.
• Please explain what was the data resolution used in this study (hourly, daily…) and why?
• For better understanding of the methodology I would recommend adding a flowchart.
• What type trees did the authors used to extract information about carbon concertation

Validity of the findings

• The result section is well written.
• Some sensitivity analysis is needed, perhaps using boxplots.
• Figure 3, the maps are a low quality, please improve and scale bare. Legend and north arrow.
The discussion is a bit naïve, please elaborate it more. The discussion should provide a summary of the main finding(s) of the manuscript in the context of the broader scientific literature, as well as addressing any limitations of the study or results that conflict with other published work.
• This section is well written, nevertheless is too short. Please include some quantitative findings and elaborate it further.
• State the main findings resulted from this work through bullet points.

Additional comments

• I would suggest to elaborate further this section by adding more literature.
• The objectives are not explicitly stated. It is not very clear what is the contributions of this work.
Please explicitly state the main objectives of this work.

---

## Round 0.2 · accepted · Accept

The reviewers agree that your manuscript has been improved and their remarks have been correctly adressed.

·

Basic reporting

The revised version of the manuscript is greatly improved.

Experimental design

I appreciate that the authors now more clearly explain the difficulties in making the eddy flux measurement. Their added text and citations makes their case a more viable, realistic-sounding approach.

Validity of the findings

I believe that the paper will not be improved significantly with much further revision.

Additional comments

See attached document for a few, additional but minor comments.

Reviewer 2 ·

Basic reporting

Well done!

Experimental design

Well done!

Validity of the findings

Well done!

Additional comments

Well done!